# The use of culturally adapted and translated depression screening questionnaires with South Asian haemodialysis patients in England

Shivani Sharma[1]*, Sam Norton[2], Kamaldeep Bhui[3], Roisin Mooney[1], Emma Caton[1], Tarun Bansal[4], Clara Day[5], Andrew Davenport[6], Neill Duncan[7], Philip A. Kalra[8], Maria Da Silva-Gane[9], Gurch Randhawa[10], Graham Warwick[11], David Wellsted[1], Magdi Yaqoob[12], Ken Farrington[9]

1 School of Life and Medical Sciences, University of Hertfordshire, Hatfield, United Kingdom, 2 Psychology Department, Institute of Psychiatry, Psychology & Neuroscience, King's College London, London, United Kingdom, 3 Department of Psychiatry, University of Oxford, Oxford, United Kingdom, 4 Renal Medicine, Bradford Teaching Hospitals, NHS Foundation Trust, Bradford, United Kingdom, 5 Renal Medicine, University Hospitals Birmingham NHS Foundation Trust, Birmingham, United Kingdom, 6 Centre for Nephrology, Division of Medicine, Royal Free and University College Hospital Medical School, London, United Kingdom, 7 Division of Renal Medicine and Transplantation, Imperial College Healthcare NHS Trust, London, United Kingdom, 8 Department of Renal Medicine, Salford Royal NHS Foundation Trust, Salford, United Kingdom, 9 Queen Elizabeth Renal Unit, Lister Hospital, East and North Hertfordshire NHS Trust, Stevenage, United Kingdom, 10 Institute for Health Research, University of Bedfordshire, Luton, United Kingdom, 11 John Walls Renal Unit, Leicester General Hospital, Leicester, United Kingdom, 12 Barts Health NHS Trust, London, United Kingdom

* s.3.sharma@herts.ac.uk

## Abstract

### Background

Depression is common amongst patients receiving haemodialysis (HD). Assessment and intervention when faced with language and cultural barriers is challenging. To support clinician decisions, we conducted a cross-sectional study to assess the use of culturally adapted and translated versions of commonly-used depression screening questionnaires with South Asian patients receiving HD in England.

### Methods

Patients completed adapted versions of the Patient Health Questionnaire (PHQ-9), the Centre for Epidemiological Studies Depression Scale Revised (CESD-R), and the Beck Depression Inventory II (BDI-II). All questionnaires were available in Gujarati, Punjabi, Urdu, and Bengali. A comparative sample of white-Europeans completed the questionnaires in English. The research was based across 9 National Health Service (NHS) Trusts in England. Structural validity of translated questionnaires was assessed using confirmatory factor analysis. Diagnostic accuracy was explored in a subgroup of South Asians against ICD-10 categories using the Clinical Interview Schedule Revised (CIS-R) with receiver operating curve (ROC) analysis.

**Data Availability Statement:** "Data supporting this publication can be obtained from the University of

Hertfordshire Research Archive (https://doi.org/10.18745/DS.26081)."

**Funding:** This work was supported by the National Institute for Health Research (NIHR; www.nihr.ac.uk) (Research for Patient Benefit programme PB-PG-1112-29078; award to SS, KB, CD, AD, ND, MDS-G, GR, GW, DW, MY, KF). The views expressed are those of the author(s) and not necessarily those of the NIHR or the Department of Health and Social Care. The funders had no role in the study design, data collection and analysis, decision to publish, or preparation of the manuscript.

**Competing interests:** The authors have declared that no competing interests exist.

**Abbreviations:** BDI-II, Beck Depression Inventory-II; CESD-R, Centre for Epidemiological Studies Depression Scale-Revised; DSM-IV, Diagnostic and Statistical Manual of Mental Disorders- IV; ESKF, End-stage kidney failure; HD, Haemodialysis; ICD-10, International Classification of Diseases-10; NICE, The National Institute for Health and Care Excellence; NHS, National Health Service; PHQ-9, Patient Health Questionaire-9; ROC, Receiver operating curve.

## Results

229 South Asian and 120 white-European HD patients participated. A single latent depression factor largely accounted for the correlations between items of the PHQ-9, CESD-R and BDI-II. Issues with measurement equivalence implied that scores on the translations may not be comparable with the English language versions. Against CIS-R based ICD-10 diagnosis of depression, sensitivity was modest across scales (50–66.7%). Specificity was higher (81.3–93.8%). Alternative screening cut-offs did not improve positive predictive values.

## Conclusions

Culturally adapted translations of depression screening questionnaires are useful to explore symptom endorsement amongst South Asian patients. However, data indicate that standard cut-off scores may not be appropriate to classify symptom severity. Use of the CIS-R algorithms for optimal case identification requires further exploration in this setting. Strategies to encourage recruitment of under-represented groups in renal research are also warranted, especially for in-depth discussions related to psychological care needs.

## Background

Data from the UK suggests that people of South Asian origin (Indian, Pakistani, Bangladeshi) are three to five times more likely to develop end-stage kidney failure (ESKF) compared to the white majority [1, 2]. One explanation for this is the higher prevalence of diabetes [3] and hypertension [4] among South Asian groups, both of which have been identified as significant risk factors for ESKF [5]. South Asians comprise around 5% of the UK population yet represent 12.2% of renal service users [6]. Diabetic nephropathy is more common and progression to ESKF often faster [7, 8]. South Asians wait on average a year longer to receive a donor organ [9], making haemodialysis (HD) a common treatment modality.

While many South Asian patients are likely to have good English language skills, others struggle and often rely on family members to translate [10]. At least 1.6% of the UK population speaks very little or no English [11]. Limited English Proficiency complicates healthcare decision-making and impairs patient satisfaction with care [12–15]. Importantly, language and cultural barriers may also hinder identification of comorbid conditions such as depression, which cluster in ESKF [16]. Depression is the most common psychological complaint amongst HD patients, affecting around one third of individuals [17]. Identification is important given its impact on quality of life and prognosis [18–21]. Detection and management in individuals from minority ethnic communities is recognised as a significant challenge [22–24].

In South Asian languages, there are no well-established linguistic equivalents for a medical diagnosis of depression [25]. This in itself may signal differing beliefs in relation to mental health. Multiple cultural factors have been implied in under-diagnosis including conflicting cultural expectations, differing explanatory models of mental health, alternative pathways to help-seeking as well as difficulties in communicating with health care providers [26–29]. As such, despite strong policy commitments, many countries with multi-ethnic demographics struggle to reduce disparity in access, experience and psychological outcomes for patients affected by long-term conditions such as ESKF [16]. This is an urgent yet understudied problem in the renal community.

The aim of this study was to culturally-adapt, translate and validate three widely used depression screening questionnaires for use with South Asian patients receiving HD in England.

## Methods

### Study setting and approvals

A cross-sectional study was conducted across nine National Health Service (NHS) HD Centres in England. Centres were selected because of higher proportions of patients being of South Asian origin (range 14.3–38.5%) [30]. Local care teams were consulted to help identify the language needs of patients. Ethical approval was provided by the Health Research Authority London Hampstead Research Ethics Committee (REC) (reference 14/LO/0568). Informed written consent was obtained from all participants at each stage of the research.

### Patient recruitment and data collection

Patients were eligible to take part in the research if (1) aged over 18 years (2) had been receiving HD for over 6 months (3) were verbally fluent in a target language (4) not currently or within the last year referred, diagnosed or treated for psychological complaints and (5) had capacity to consent. Those with existing psychiatric diagnoses, cognitive impairment, and developmental delays were excluded.

Recruitment to the study took place between June 2015 and February 2017. Patients were approached about the study during visits for in-centre HD. They were provided with a verbal explanation about the study alongside written information in English, their language of origin and the roman script of the same. Effort was made for South Asian patients to be approached by a member of the care team with suitable linguistic skills and/or to engage carers accompanying patients to help explain the study. Those interested were contacted by a bilingual project worker to explain the research further and to arrange participation. The project worker either handed patients the questionnaires to self-complete or supported them by reading out items and noting answers. Diagnostic interviews were conducted by the same project workers in a private room either before or after dialysis.

All patients completed three depression screening tools: the Patient Health Questionnaire (PHQ-9) [31], Centre for Epidemiological Studies Depression Scale Revised (CESD-R) [32], and the Beck Depression Inventory © NCS Pearson Inc Reproduced with Permission (BDI-II) [33]. We also assessed the Whooley Questions [34] recommended by the National Institute for Health and Care Excellence (NICE) as a brief screen for depression. For South Asian patients, these questionnaires were adapted and made available in Gujarati, Punjabi, Urdu and Bengali. Any patient who scored above 0 on a questionnaire item related to self-harm was followed-up by a Consultant Nephrologist within 24 hours of the disclosure. In this case, the research team will have shared the patients identity with the local study lead. Patient participants were made aware of this for safeguarding when consenting to join the study. All patients were offered the opportunity to take part in a follow-up diagnostic interview using the Clinical Interview Schedule Revised (CIS-R). A comparative sample of white-European English-speaking patients completed the depression screening measures only.

### Data collection measures

**PHQ-9 [31].** The PHQ-9 is a 9-item screening questionnaire for major depressive disorder. Items are scored using a four-point ordinal scale from 0 (not at all) to 3 (nearly every day). Total scores indicate the severity of depressive symptoms (range 0 to 27) and can be allocated

to one of five severity groupings. Scores ≥10 are considered to have screened positive for probable major depressive disorder. The PHQ-9 has good psychometric properties in patients with ESKF [35]. The UK NICE guidelines for depression [36] suggest that an almost identical dichotomous version of the first two items of PHQ-9 can be used for case identification in high risk groups, such as those with long-term conditions. These items, referred to as the 'Whooley Questions' [34], use the binary yes/no format and offer a briefer screening instrument. The Whooley Questions were also included.

**CESD-R [32].** The CESD-R is a 20-item screening questionnaire with a revised scoring system enabling classification of the 20 answers in terms of DSM-IV diagnostic criteria. Items are scored using a five-point ordinal scale from 0 (not at all) to 4 (nearly every day for 2 weeks). The total score indicates the severity of depressive symptoms. A score ≥16 is commonly used to indicate probable depression 'caseness'. The CESD-R has been used with patients with ESKF [37].

**BDI-II [33].** The BDI-II comprises 21 items rated on a 4-point ordinal scale (0–3) indicating the frequency or severity of a particular depressive symptom. The total score (range 0 to 63) indicates the severity of depressive symptoms. The psychometric properties of the BDI-II in dialysis patients have been shown to be acceptable [38]. A score ≥16 compares favourably to diagnostic measures among UK HD patients [38].

**CIS-R [39].** The CIS-R is a structured assessment for identifying cases of ICD-10 diagnoses of major depressive disorder. It can be used by lay, trained interviewers. We considered 10 symptom domains and utilised the classification systems described by Singleton, Lee and Meltzer [40], which classifies CIS-R scores into ICD-10 diagnoses with three levels of severity: mild, moderate, and severe. For the mild and moderate categories, it is possible to distinguish between people with and without somatic complaints. For those in the severe condition, somatic complaints must be present. The construct validity of the CIS-R has previously been demonstrated to be valid cross-culturally with minority ethnic groups in the UK general population [41].

## Cultural adaptation and translation procedure

Translation of the depression screening questionnaires and the CIS-R followed the framework for the Translation and Cultural Adaptation Process for Patient Reported Outcomes [42]. Table 1 summarises the key stages of questionnaire adaptation and on-going quality assurance throughout the study. Importantly, this framework has an emphasis on conceptual equivalence, which the stages of simply translating measures do not explicitly attend to. Conceptual equivalence is essential to ensure that adapted measures can be meaningfully interpreted in different cultures. There was an emphasis throughout the process on use of the same terms where this degree of harmonisation was sensible. For example, appetite was given as bhukh; sazza for feelings of punishment, though spelling varied in the different language groups to reflect pronunciation.

## Recruitment and training of bilingual project workers

Twelve bilingual project workers of South Asian heritage supported patient recruitment, consent, questionnaire and interview completion. All had experience of working with patients in sensitive contexts. They were recruited through involvement in existing projects (n = 6) with minority ethnic patients or through the NHS Trusts where the research was based. Here they were employed in roles such as dieticians and research nurses (n = 6). A comprehensive schedule of training and on-going support was put in place (detailed in supplementary material).

**Table 1. Summary of adaptation and translation of questionnaire and diagnostic interview.**

| Cultural Adaptation Process for Patient-Reported Outcomes | |
| --- | --- |
| *Preparation* | Permission sought for the adaptation of the BDI-II from Pearson Publishing (not in the public domain) |
| | Project team produced information about the conceptual basis of all questionnaires and the CIS-R for use in training those involved in measure adaptation and translation. This included a review of any existing, freely available translations (applied to PHQ-9 only) |
| | Recruitment of 3 translators per language through professional agencies and existing contacts from previous research studies with South Asian HD patients |
| *Translation* | Training in conceptual basis of measures and study rationale. Encouraged to use lay language that would be easily understood by the majority of people |
| | Focused on conceptual and not literal equivalence |
| | Preliminary discussion about any difficult items to agree solutions |
| | Two independent forward translations of each screening questionnaire and the CIS-R |
| *Reconciliation* | Across a series of meetings, forward translations reconciled to reach consensus |
| *Back-translation* | Third translator reviewed reconciled versions of screening questionnaires and the CIS-R with a focus on spelling, grammar, comprehension, language complexity |
| | Independent back translations of all screening questionnaires and the CIS-R (via professional services unrelated to the study) |
| *Review of back translation* | First author reviewed the back translations and resolved any discrepancies |
| *Harmonisation* | Series of panel meetings with all involved in translation development to harmonise across languages and with English language source |
| | Where similar words existed, care taken for consistency e.g. the BDI-II item on 'punishment' equated to the word 'sazza' in both Punjabi and Urdu, 'saja' in Gujarati and 'shaja' in Bengali |
| *Cognitive de-briefing* | Harmonised translations tested in focus groups including 5–8 people (1 per language group) involving lay community members who were native speakers of one of the target languages |
| *Finalisation* | Feedback from focus groups included in finalised version of screening questionnaires and CIS-R |
| *Quality Assurance* | All screening data entered online to monitor risk throughout the study e.g. suicidal ideation and checked against hard copy |
| | All CIS-R audio interviews independently checked for accuracy |

BDI-II = Beck Depression Inventory-II; PHQ-9 = Patient Health Questioannire-9; CIS-R = Clinical Interview Schedule Revised

## Sample size justification

Sample sizes for screening and diagnostic interviews were determined a priori. We aimed to recruit 250 South Asian patients to complete depression screening questionnaires, approximately 50 per language group. Target sample sizes based on the precision with which parameters (e.g., loadings) are estimated and likely recovery of the correct factor structure are recommended [43]. In addition to sample size, this is influenced by the number of factors, number of items per-factor, and expected item communalities. Around 50 per language group is appropriate based on published simulation studies indicating minimum sample sizes of 18 [44] and 33 [43] being sufficient for unidimensional scales with at least 9 items per-factor and high communalities as seen with the scales considered. Furthermore, a simulation study of measurement invariance testing indicated sample size of 200 is sufficient to achieve over 80% power where there are few factors and high communalities [45]. We further aimed to follow-up a subset of 150 South Asian patients with a diagnostic interview, where a sample size of 150 would provide acceptable precision for estimating overall diagnostic accuracy–i.e. a maximum

standard error of 4.1%, by the exact method [46], and 95% confidence intervals with a maximum width of approximately +8.0%.

## Statistical analysis

Item response modelling employed the graded response model, estimating parameters using full information (robust) maximum likelihood in a latent variable modelling framework using Mplus version 7.4. Given the polytomous response distribution for each of the items these were specified as ordered categorical variables in Mplus. Although estimated as a form of confirmatory factor analysis, this is equivalent to a graded response model, where factor loadings are a transformation of the discrimination parameter for each item and thresholds a transformation of the difficulty parameter for each level of response for the item [47]. The discrimination parameter indicates the correlation between the item response and the underlying latent depression construct. The difficulty parameter indicates the level of the latent variable where there is a 50% chance of responding positively to that item. Sufficient unidimensionality of the depression screening questionnaires was assessed by examining the proportion of the variance explained by the first principle component of the inter-item polychoric correlation matrix and applying parallel analysis to determine whether one or more underlying latent constructs is likely to explain the pattern of item responses. Since maximum likelihood estimation is used standard model fit statistics are not provided.

We considered whether the assumption of measurement equivalence held by testing hypotheses relating to configural and scalar invariance. Respectively, if the first hypothesis is met then the scales can be said to measure the same underlying construct (i.e. depression) whereas the second must be met for the comparisons of means across groups to be considered valid–that is, the scales may measure depression but the same scores on the scale across translations may not indicate the same severity of depressive symptoms.

Configural invariance was assessed by determining whether the intended factor structure provided an acceptable fit across all models. Since the intended models all involved a unidimensional depression factor, configural invariance was assumed to hold where parallel analysis indicated a unidimensional solution and where all items had a significant discrimination parameter in the graded response model.

Scalar invariance was assessed using likelihood ratio tests to compare non-equivalent and scalar equivalent models for each of the depression questionnaires [48, 49]. The non-equivalent model estimates separate factor loadings and threshold parameters for each language, whereas the scalar equivalence model constrains these parameters to be the same across languages. Scalar equivalence must hold for comparisons of scores across languages to be appropriate. It should be noted that the small number of individuals within each language group means that power for the likelihood ratio test of scalar equivalence is relatively low. Therefore, the significance level was set at 10% and the analysis were considered exploratory rather than confirmatory (i.e. a significant result indicates need for further research). In Mplus this was achieved using multigroup confirmatory factor analysis with the scalar model subcommand.

Diagnostic accuracy for major depression was explored against the CIS-R for a subgroup of individuals where this was available. Sensitivity, specificity and other information relating to diagnostic accuracy was calculated using the recommended clinical cut-offs for each scale. Receiver operator characteristic curves (ROC) were calculated to determine the diagnostic accuracy of the scales across the full range of values.

## Patient and public involvement

Patient and public involvement was embedded in the research process through a project advisory group. This group included a South Asian patient with experience of both a renal

transplant and HD; a South Asian patient with diabetes; a GP located in an area with high representation from South Asians; and a Psychiatrist with expertise in working with multi-ethnic groups. Patient members specifically helped shape the research questions, recruitment strategy, and methods of dissemination. They actively contributed to the protocol for the research including writing the lay summary. The advisory group oversaw the conduct of the study, acting as 'critical friends' throughout.

## Results

### Participant characteristics

Two-hundred and fifty South Asian patients were approached; 236 consented to participate; 229 returned completed questionnaires. All 229 patients completed three depression screening tools. Only 18% of patients were able to complete the questionnaires unassisted ($n = 42$). Sixty patients consented for interview. A comparative sample of 120 white English-speaking patients (58% consent rate) also completed the depression screening questionnaires.

Overall the sample reflected a typical UK HD patient population except for the deliberately high proportion of South Asian ethnicity (Table 2). There were some differences between South Asian and white English-speaking groups, notably the South Asians had a lower mean age (63.5 versus 68.5, t(347) = -3.35, p = 0.001), more women (44% versus 31%, z = 2.41, p = 0.016), more had diabetes (56% versus 33%, z = 4.01, p < 0.001) and dialysis Kt/V was higher (1.67 versus 1.47, t(347) = 4.17, p < 0.001).

### Screening questionnaire results

Table 3 includes data from the screening questionnaires. In general, the scores in the South Asian group and proportions of screening test scores above the cut-off value were higher than in the white English-speaking group. This was mainly driven by high scores in the Urdu group.

### Validation

Item response patterns for the South Asian groups are provided in the supplementary material. Acceptability of the scales was good as indicated by item completion rates–the highest non-completion rate was 16% (n = 37; 95%CI 12% to 22%) for the BDI-II item on sex. The

**Table 2. Demographic and clinical characteristics.** Continuous variables expressed as mean ± standard deviation or median (interquartile range) as appropriate. Categorical variables expressed as percentages. *p = 0.016, **p = 0.001, ††p <0.001.

|  | Gujarati (N = 44) | Punjabi (N = 67) | Urdu (N = 72) | Bengali (N = 46) | Total South Asian (N = 229) | English (N = 120) |
|---|---|---|---|---|---|---|
| **Age (years)** | 63.8 ± 14.3 | 64.7 ± 10.6 | 61.2 ± 14.3 | 64.9 ± 13.4 | 63.5 ± 13.1** | 68.5 ± 13.5 |
| **Female (%)** | 50 | 28 | 49 | 54 | 44* | 31 |
| **Born outside UK (%)** | 93 | 91 | 92 | 100 | 93 | 0 |
| **Dialysis Vintage (months)** | 37.8 (65.2) | 43.4 (61.9) | 30.2 (53.6) | 33.5 (15) | 33.6 (50) | 36.7 (41.1) |
| **Diabetes (%)** | 48 | 56 | 54 | 67 | 56†† | 33 |
| **Charlson comorbidity Index** | 5.3 ± 2.1 | 5.7 ± 1.6 | 5.9 ± 2.3 | 5.9 ± 2.3 | 5.7 ± 2.0 | 6.1 ± 2.5 |
| **Albumin (g/L)** | 38.2 ± 3.6 | 34.7 ± 4.8 | 35.8 ± 4.1 | 38.9 ± 6.5 | 36.6 ± 5.1 | 37.3 ± 5.0 |
| **Haemoglobin (g/L)** | 109 ± 14 | 112 ± 16 | 111 ± 15 | 111 ± 14 | 111 ± 15 | 110 ± 12 |
| **Kt/V** | 1.64 ± 0.50 | 1.68 ± 0.47 | 1.62 ± 0.48 | 1.78 ± 0.39 | 1.67 ± 0.46†† | 1.47 ± 0.35 |

Kt/v = measure of treatment adequacy.

**Table 3. Median (interquartile range) scores across depression screening questionnaires and proportion of patients scoring above established screening thresholds.**

| | Gujarati (N = 44) | | Punjabi (N = 67) | | Urdu (N = 72) | | Bengali (N = 46) | | Total South Asian (N = 229) | | English (N = 120) | |
|---|---|---|---|---|---|---|---|---|---|---|---|---|
| | **A. Median values of total scores (interquartile range)** | | | | | | | | | | | |
| **PHQ-9** | 4 | (5.75) | 5 | (11) | 11 | (12.5) | 3 | (5.25) | 6 | (10) | 5 | (7.75) |
| **CESD-R** | 9 | (13.75) | 4 | (32) | 21 | (39) | 6 | (13) | 11 | (24.25) | 9 | (17) |
| **BDI-II** | 9 | (11) | 10 | (25) | 16 | (27) | 4.5 | (12.75) | 10 | (20.25) | 8.5 | (9.75) |
| **Whooley** | 0 | (1) | 0 | (2) | 1 | (2) | 0 | (1) | 0 | (2)* | 0 | (1.75) |
| | **B. Number (proportion) with scores > cut-off for high probability of Major Depressive Disorder** | | | | | | | | | | | |
| **PHQ-9 ≥ 10** | 6 (13.6%) | | 24 (35.8%) | | 40 (55.6%) | | 8 (17.4%) | | 78 (34.1%) | | 30 (25%) | |
| **CESD-R ≥ 16** | 12 (27.3%) | | 26 (38.8%) | | 45 (62.5%) | | 12 (26.1%) | | 95 (45.5%)** | | 45 (37.5%) | |
| **BDI-II ≥ 16** | 9 (25.5%) | | 25 (37.3%) | | 35 (50.7%) | | 9 (19.6%) | | 78 (35.1%)†† | | 28 (23.3%) | |
| **Whooley ≥1** | 14 (31.8%) | | 29 (43.3%) | | 45 (62.5%) | | 17 (37.0%) | | 105 (45.9%) | | 53 (44.2%) | |

For comparison of South Asians vs English

*p = 0.042

** p = 0.035

†† p = 0.024 (by Mann-Whitney U test). For comparison between South Asian groups p <0.01 for all screening tests (by Kruskal- Wallis test)

distribution of the total scores was positively skewed with many patients scoring at the minimum across all scales.

Measurement equivalence was assessed for the PHQ-9, CESD-R and BDI-II, the results of which are given in Table 4. Structural validity in terms of configural invariance for each of the scales was good with each matching the intended factor structure in the overall sample–the first principal component explained approximately two-thirds of item response variability across all scales in all language subgroups (Table 4). Parallel analysis confirmed that for each of the scales in each of the South Asian and English Language groups that a single underlying latent factor was sufficient to account for the correlations between items. Together these results suggest that the scales function sufficiently well in each of the South Asian samples to be used as a measure of depressive symptoms. Reliability of the total scores for each scale were estimated using Chronbach's alpha (Table 4). The minimum levels of alpha for each scale were .74 for the PHQ-9, .85 for the CESD-R, and .91 for the BDI-II.

Additional analysis considered whether the assumption of scalar invariance was met (Table 4). Tests using multiple group confirmatory factor analysis indicated that, compared to the English language sample, each of the screening instruments failed to meet this assumption for at least one of the groups (as indicated by a significant p-value). While each of the scales

**Table 4. Measurement equivalence of translated depression screening instruments.**

| | | Gujarati (N = 44) | Punjabi (N = 67) | Urdu (N = 71) | Bengali (N = 46) | Total South Asian (N = 228) | English (N = 120) |
|---|---|---|---|---|---|---|---|
| **PHQ-9** | **PC1 variance explained** | 50% | 71% | 59% | 50% | 63% | 59% |
| | **Chronbach's alpha** | 0.741 | 0.902 | 0.86 | 0.769 | 0.87 | 0.838 |
| | **Scalar invariance vs English** | p < .001 | p = .001 | p = .080 | p = .280 | p < .070 | |
| **CESD-R** | **PC1 variance explained** | 45% | 59% | 52% | 47% | 64% | 48% |
| | **Chronbach's alpha** | 0.851 | 0.959 | 0.943 | 0.867 | 0.943 | 0.926 |
| | **Scalar invariance vs English** | p < .001 | p < .001 | p = .230 | p < .001 | p < .055 | |
| **BDI-II** | **PC1 variance explained** | 47% | 54% | 52% | 46% | 65% | 54% |
| | **Chronbach's alpha** | 0.914 | 0.954 | 0.945 | 0.931 | 0.945 | 0.908 |
| | **Scalar invariance vs English** | p < .001 | p = .006 | p < .001 | p < .001 | p < .001 | |

**Table 5. Exploratory analyses of diagnostic accuracy (N = 50).**

|  | PHQ9≥10 | CESD-R≥16 | BDI-II≥16 | Whooley |
|---|---|---|---|---|
| Screen positive | 22% (11.5% to 36.0%) | 28% (16.2% to 42.5%) | 28% (16.2% to 42.5%) | 36% (22.9% to 50.8%) |
| Sensitivity | 50.0% (26.0% to 74.0%) | 52.9% (27.8% to 77.0%) | 61.1% (35.7% to 82.7%) | 66.7% (41.0% to 86.7%) |
| Specificity | 93.8% (79.2% to 99.2%) | 84.4% (67.2% to 94.7%) | 90.6% (75.0% to 98.0%) | 81.3% (63.6% to 92.8%) |
| ROC area | 71.9% (59.3% to 84.5%) | 68.7% (54.9% to 82.5%) | 75.9% (63.2% to 88.5%) | 69.4% (56.4% to 82.4%) |
| Positive predictive value | 81.8% (48.2% to 97.7%) | 64.3% (35.1% to 87.2%) | 78.6% (49.2% to 95.3%) | 66.7% (41.0% to 86.7%) |
| Negative predictive value | 76.9% (60.7% to 88.9%) | 77.1% (59.9% to 89.6%) | 80.6% (64.0% to 91.8%) | 81.3% (63.6% to 92.8%) |
| Likelihood ratio (+) | 8.0 (1.9 to 33.1) | 3.4 (1.4 to 8.5) | 6.5 (2.1 to 20.4) | 3.6 (1.6 to 7.9) |
| Likelihood ratio (-) | 0.5 (0.3 to 0.9) | 0.6 (0.3 to 0.9) | 0.4 (0.2 to 0.8) | 0.4 (0.2 to 0.8) |
| Diagnostic Odds ratio | 15.0 (3.0 to 356.3) | 6.1 (1.6 to 22.6) | 15.2 (3.5 to 64.9) | 8.7 (2.4 to 31.8) |

appear to measure the same underlying depression construct, the scores on the scales may not be comparable across the different translations since they may indicate different levels of symptom severity. Inspection of individual items did not indicate specific items that accounted for these findings, rather it would appear that an accumulation of minor differences across items perturbs the total scale scores.

## CIS-R diagnostic interview data

Of the 229 South Asian patients, a sub sample of 60 individuals also completed the CIS-R diagnostic interview; sufficient data were available for 50 (83%) individuals. The largest patient sub-group was Gujarati speaking (n = 20) followed by Bengali (n = 18), Urdu (n = 8), and Punjabi (n = 4). In total, 18 (36%) of the 50 patients met the CIS-R criteria for a current depressive episode. Of these, ten (20%) were classified as experiencing a moderate depressive episode and 8 (16%) were classified as having a severe depressive episode.

We progressed with exploratory tests of diagnostic accuracy relative to the CIS-R ICD-10 diagnosis of depression (mild upward) for the PHQ-9, CESD-R and BDI-II and Whooley items categorising patients as screening positive at previously defined cut-points for each of the tools (Table 5). Specificity for all tools was relatively high (81.3% to 93.8%), however, sensitivity was low (50% to 66.7%). The true positive rate ranged between 64.3% and 81.8% and the false positive rate ranged between 76.9% and 81.3%. Because of the small sample size, little inference can be drawn about the comparative accuracy between screening tools.

ROC curves were used to further explore the accuracy of the PHQ-9, CESD-R and BDI-II across the full range of each tool, with respect to CIS-R diagnostic grouping (Fig 1). The area under the ROC curves, which is equivalent to the probability that an individual selected at

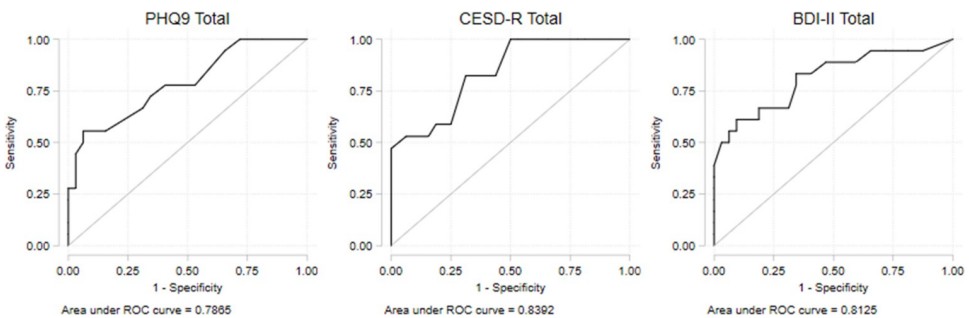

**Fig 1. Receiver Operating Characteristic (ROC) curves depicting the accuracy of the PHQ-9, CESD-R and BDI-II across the full range of each tool with respect to CIS-R diagnostic grouping, in 50 South Asian HD patients.**

random is correctly classified by CIS-R diagnosis, was similar for each tool ranging between .79 and .84. There was no indication that diagnostic validity would be increased using alternative cut-offs to the suggested cut-offs for the scales assessed above.

## Discussion

To our knowledge, this is the first study to exclusively prioritise the measurement of depression amongst patients of minority ethnic heritage receiving HD in England. This is a significant step forward in improving access to psychological care. Screening tools are usually an important part of identifying patients who would benefit from support but we know little about their use in this patient population [16].

Language and/or cultural barriers can hinder recruitment into research, resulting in potential biases in relation to whom the outcomes represents and benefit. This research was designed to overcome such challenges. We focused on South Asian patients given the disproportionate burden of ESKF in these communities [1, 2].

Our findings indicate that patients engaged well with the adapted screening questionnaires. Ninety-six per cent of those who consented went on to participate. Consent rates were notably more favourable than in a comparative sample of white-Europeans (58%). This likely reflects the benefit of using bilingual project workers throughout the research, especially given the high proportion of South Asians who required support in completing the questionnaires due to limited literacy in their language of origin. Favourable recruitment might also signal less research 'fatigue' in under-represented patient groups, though we would need to explore this assertion further by noting reasons for refusal. Data completeness was excellent across individual items (see supplementary files). Only the BDI-II question on 'interest in sex' was problematic. This is unsurprising as sex is a taboo topic in South Asian communities [50]. Interventions that increase awareness of the positive aspects of sex may be advantageous to address this. The overall pattern of responses was otherwise in line with that expected in a HD patient population; providing some support for item comprehension.

The scales reliably measured the underlying construct of depression. Across all questionnaires, between 34 and 46% of South Asian patients were identified as experiencing significant depressive symptoms (Table 2). In descriptive terms, this is higher than the proportion of patients in the English-language group (25–44%). However, since the property of scalar equivalence did not hold, scores on the translated versions are unlikely to be comparable with the English language scales. It is probable that there are differences in how patients experience, appraise and express depressive symptoms. This does not mean that the scales themselves are problematic. It signals that between group comparisons are not suitable [51] and may only render approximate results as the underlying latent response functions are not identical. Previous research has shown that South Asian patients receiving HD have higher self-reported symptom burden [52]. South Asians and people with depression or medical disorders are each more likely to have somatic symptoms, and all three of these characteristics may mask somatised distress [41]. Biases in how patients endorse specific items may introduce spurious differences, which impact on the interpretation of meaning of overall symptom levels. We took significant steps to ensure that the adaptation and translation process was optimal, so it is unlikely that these issues explain the findings.

Analyses related to diagnostic accuracy were exploratory. Only 26% of South Asian patients went on to complete follow-up interviews, with data completeness for 83%. This is considerably lower than our target sample of 150 patients engaging at the stage. A minority of patients provided reasons for opting out. Where available, this related to logistic issues or advice from family members not to engage. Strategies to overcome such barriers are needed and may

include more flexibility in interview completion such as via telephone as well as engaging caregivers in understanding the value and usefulness of research. A recent study involving South Asian caregivers of patients with Dementia also reported that factors such as confusion over research and mistrust are significant obstacles to participation [53]. We progressed with exploratory analyses nonetheless as there is no data on the diagnostic accuracy of adapted screening questionnaires with the South Asian HD patient population. Given that screening questionnaires play a crucial role in triggering further psychological care, it is important to explore the extent to which they may concur with more robust case identification.

The proportion of HD patients diagnosed with depression identified using the CIS-R (36%) was much higher than that found in other HD populations (19–22%) using structured diagnostic interviews [35, 38]. This likely reflects the sampling method. The area under the ROC curve for each instrument ranged between .79 and .84. A review for NICE [36] on diagnostic accuracy of depression screening tools reported levels between .79 and .95. More importantly, sensitivity was relatively low for all tools in our study implying that a significant proportion of patients with depression may be missed. This is unlikely to reflect an issue with scales themselves since they correlated highly with each other and the symptom profiles were typical of HD populations.

Gholizadeh et al [54] looked at the screening potential of an Urdu version of the PHQ-9 in patients with coronary artery disease against the depressive episode module of the Mini International Neuropsychiatric Interview. Their study was based in Islamabad, Pakistan. The sensitivity of the PHQ-9 at $\geq 6$ was 76%. This is much higher than for any screening instrument in the current study and notably with a lower screening threshold. A key difference in this study was administration of the diagnostic interview via a trained clinician over lay researcher alongside a larger sample of patients. To help clarify anomalies in our data, we suggest that a suitably powered study is needed in which there are broader strategies to increase patient recruitment to interview. We also advance administration of the CIS-R or other diagnostic scheme via a clinician such as a psychiatrist or psychologist so that the nuances of symptom overlap between depression and ESKF can also be teased out. An alternative would be to reduce emphasis on diagnostic labelling per say and to work with patients to understand what matters most to them to improve outcomes [55]. The latter is a more pragmatic approach given the cost implications of recruiting a sufficient sample and practicability of offering culturally tailored diagnosis in clinical settings.

In interpreting our findings, there are of course some caveats. The overall sample size (N = 229) was small for evaluating factor models in each language sub-group. To our advantage, these scales have a long history of evaluation which demonstrates a robust underlying construct which has, in this data, been well modelled by a single factor, which limits the number of parameters required to estimate. In addition, the translated scales did fit the assumed model well with $\alpha > .85$ for the CESD-R and BDI-II and in excess of .74 for the PHQ-9. The main limitation of the smaller sample size has been in determining the equivalence of symptom severity (e.g. scaler invariance) between the translated scales, where a larger sample size may have provided more information about the nature of these differences. The sample size for the evaluation of diagnostic accuracy was also insufficient for an evaluation of this kind. This has inevitably led to wide confidence intervals for the parameters reported in Table 5, and for the ROC functions estimated to demonstrate the reliability of the tests to evaluate the presence of diagnosable cases. Our parameter estimates are, however, broadly in line with previously reported estimates. Completion of the CIS-R was also not based on random sampling, and so the prevalence rate is likely to be biased and the true positive and negative rates unlikely to generalise to the wider population. Finally, there were some important differences between the South Asian and White heritage comparator groups, which may have impacted the proportion of patients scoring above established cut-off i.e. presence of comorbidity. Notwithstanding

these limitations, this is the first study in HD centres in England to provide access to language of origin based screening. We have demonstrated robustness in cultural tailoring from measure adaptation through to recruitment and participation. Our data are amongst few studies, as noted in NICE guidelines [34], to take such an approach to considering essential steps in the pathway to psychological care for minority ethnic patients.

## Conclusions

Culturally adapted and translated screening questionnaires are useful to initiate clinical enquiry about the meaning of depressive symptoms. However, our data do not support their use to classify symptom severity in South Asian patients based on established thresholds. Due to limited recruitment to diagnostic interviews, our data on diagnostic accuracy are nuanced. A larger study is required to address the optimal cut points to increase sensitivity across individual scales. A patient-centred, less label bound approach is also advocated as an alternative and perhaps more pragmatic approach to triggering psychological intervention. Finally, we suggest that researchers aiming for meaningful cultural tailoring consider the use of bilingual project workers given low language of origin literacy in this patient population, which our study is the first to attempt to approximate.

## Supporting information

**S1 Fig. Study flow diagram.**
(PDF)

**S2 Fig.** a. PHQ-9 item response patterns. b. CESD-R item response patterns. c. BDI-II item response patterns.
(PDF)

**S3 Fig. Total score response distributions by language group.**
(PDF)

**S1 Table. Schedule of bilingual project worker training.**
(PDF)

**S2 Table.** a. PHQ-9 factor loadings from the scalar invariant model. b. CESD-R factor loadings from the scalar invariant model. c. BDI-II factor loadings from the scalar invariant model.
(PDF)

**S1 File.**
(PDF)

**S2 File.**
(PDF)

**S3 File.**
(PDF)

**S4 File.**
(PDF)

## Acknowledgments

We would like to thank all patients who took time to contribute to this research alongside the team of bilingual project workers who helped deliver the study. We are grateful to our project

advisory group, particularly patient members (Kirit Modi and Vina Nayar) for their contribution in shaping the research questions and process, and oversight of study conduct. Finally, we acknowledge the helpful feedback on statistical analyses from Joerg Schultz.

## Author Contributions

**Conceptualization:** Shivani Sharma, Kamaldeep Bhui, Andrew Davenport, Maria Da Silva-Gane, Gurch Randhawa, David Wellsted, Ken Farrington.

**Formal analysis:** Sam Norton.

**Investigation:** Roisin Mooney.

**Methodology:** Shivani Sharma, Kamaldeep Bhui, Andrew Davenport, Maria Da Silva-Gane, Gurch Randhawa, David Wellsted, Ken Farrington.

**Resources:** Shivani Sharma, Kamaldeep Bhui, Roisin Mooney, Maria Da Silva-Gane.

**Supervision:** Tarun Bansal, Clara Day, Andrew Davenport, Neill Duncan, Philip A. Kalra, Graham Warwick, Magdi Yaqoob, Ken Farrington.

**Writing – original draft:** Shivani Sharma, Sam Norton, Kamaldeep Bhui, Ken Farrington.

**Writing – review & editing:** Shivani Sharma, Sam Norton, Kamaldeep Bhui, Roisin Mooney, Emma Caton, Tarun Bansal, Clara Day, Andrew Davenport, Neill Duncan, Philip A. Kalra, Maria Da Silva-Gane, Gurch Randhawa, Graham Warwick, David Wellsted, Magdi Yaqoob, Ken Farrington.

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
