## [Decision Letter · Decision Letter 0]

18 Jan 2023

PONE-D-22-27395The use of culturally adapted and translated depression screening questionnaires with South Asian haemodialysis patients in EnglandPLOS ONE

Kia ora Dr. Sharma,

Thank you for submitting your manuscript to PLOS ONE. After careful consideration, we feel that it has merit but does not fully meet PLOS ONE’s publication criteria as it currently stands. Therefore, we invite you to submit a revised version of the manuscript that addresses the points raised during the review process.

Please consider whether you can present a formal power calculation instead of "rule of thumb" (line 229 et seq). The reviewer suggests using Cohens formula. If you can do this, that would be fine. Alternatively, you could explain that there is no universally accepted method for carrying out power calculations in factor analysis. 

Ngā mihi nui (thank you very much),

David Iain McBride

Academic Editor

PLOS ONE

Journal Requirements:

Additional Editor Comments:

Thank you for this interesting exploratory study. I apologize for the delay; however, the review is now attached. Could you please assess the comments carefully and respond as you feel appropriate.

Reviewers' comments:

Reviewer's Responses to Questions

**Comments to the Author**

1. Is the manuscript technically sound, and do the data support the conclusions?

Reviewer #1: Yes

2. Has the statistical analysis been performed appropriately and rigorously? 

Reviewer #1: Yes

3. Have the authors made all data underlying the findings in their manuscript fully available?

Reviewer #1: Yes

4. Is the manuscript presented in an intelligible fashion and written in standard English?

Reviewer #1: Yes

5. Review Comments to the Author

Reviewer #1: • The authors could present the reasons for high rates of chronic kidney disease in the minor etnicities in the UK.

• The last paragraph of the introduction could be re-written. There is no need to mention CIS-R in the introduction. The authors could move it to the methods.

• Although the authors tried to explain, a power analysis with Cohen’s formula would be more appropriate.

• The statistical analyses and the methods are appropriate.

• The authors discussed their findings with previous studies in a logical way.

Best regards

6. PLOS authors have the option to publish the peer review history of their article (what does this mean?). If published, this will include your full peer review and any attached files.

Reviewer #1: No

---

## [Author Response · Author response to Decision Letter 0]

23 Feb 2023

Dear Reviewers, 

Thank you for considering this manuscript. We appreciate your positive feedback and recommendations for improvement. Please find our point-by-point response to reviewers comments below. 

Editors Comments:

1. Please consider whether you can present a formal power calculation instead of "rule of thumb" (line 229 et seq). The reviewer suggests using Cohens formula. If you can do this, that would be fine. Alternatively, you could explain that there is no universally accepted method for carrying out power calculations in factor analysis. 

Thank you for your comment. We agree that there is no universally accepted method for calculating power for factor analytic methods. We have amended the manuscript to expand the description and the rationale for using a rule of thumb to determine sample size. The relevant text within the Methods (pg. 13) now reads: 

Target sample sizes based on the precision with which parameters (e.g., loadings) are estimated and likely recovery of the correct factor structure are recommended [43]. In addition to sample size, this is influenced by the number of factors, number of items per-factor, and expected item communalities. Around 50 per language group is appropriate based on published simulation studies indicating minimum sample sizes of 18 [44] and 33 [43] being sufficient for unidimensional scales with at least 9 items per-factor and high communalities as seen with the scales considered. Furthermore, a simulation study of measurement invariance testing indicated sample size of 200 is sufficient to achieve over 80% power where there are few factors and high communalities [45].

Thank you for this interesting exploratory study. I apologize for the delay; however, the review is now attached. Could you please assess the comments carefully and respond as you feel appropriate.

Thank you for your interest in this manuscript. Please find the reviewers comments addressed below. 

Reviewer Comments:

2. The authors could present the reasons for high rates of chronic kidney disease in the minor ethnicities in the UK.

Thank you for the suggestion. We agree that it would be useful to provide an explanation for the high rates of chronic kidney disease in patients who identify with a minority ethnic background, particularly those of South Asian heritage. As such, the following text has been added to the first paragraph in the Background (pg. 4): 

One explanation for this is the higher prevalence of diabetes [3] and hypertension [4] among South Asian groups, both of which have been identified as significant risk factors for ESKF [5].

3. The last paragraph of the introduction could be re-written. There is no need to mention CIS-R in the introduction. The authors could move it to the methods.

Following your recommendation, we have re-written the final paragraph of the introduction (pg. 6) to read as follows: 

The aim of this study was to culturally-adapt, translate and validate three widely used depression screening questionnaires for use with South Asian patients receiving HD in England. 

4. Although the authors tried to explain, a power analysis with Cohen’s formula would be more appropriate.

As mentioned above in response to the Editor, we have amended the text justifying the sample size to further explain our rationale for using a rule of thumb. The relevant text within the Methods (pg. 13) now reads: 

Target sample sizes based on the precision with which parameters (e.g., loadings) are estimated and likely recovery of the correct factor structure are recommended [43]. In addition to sample size, this is influenced by the number of factors, number of items per-factor, and expected item communalities. Around 50 per language group is appropriate based on published simulation studies indicating minimum sample sizes of 18 [44] and 33 [43] being sufficient for unidimensional scales with at least 9 items per-factor and high communalities as seen with the scales considered. Furthermore, a simulation study of measurement invariance testing indicated sample size of 200 is sufficient to achieve over 80% power where there are few factors and high communalities [45].

The statistical analyses and the methods are appropriate.

The authors discussed their findings with previous studies in a logical way.

Thank you for the positive feedback. We appreciate your support and your suggestions to help improve this manuscript.

---

## [Editor Report · Decision Letter 1]

24 Mar 2023

The use of culturally adapted and translated depression screening questionnaires with South Asian haemodialysis patients in England

PONE-D-22-27395R1

Dear Dr. Sharma,

We’re pleased to inform you that your manuscript has been judged scientifically suitable for publication and will be formally accepted for publication once it meets all outstanding technical requirements.

Kind regards,

David Iain McBride

Academic Editor

PLOS ONE

Additional Editor Comments (optional):

Thank you for undertaking those careful revisions.
---

## [Editor Report · Acceptance letter]

30 Mar 2023

PONE-D-22-27395R1 

The use of culturally adapted and translated depression screening questionnaires with South Asian haemodialysis patients in England 

Dear Dr. Sharma:

I'm pleased to inform you that your manuscript has been deemed suitable for publication in PLOS ONE. Congratulations! Your manuscript is now with our production department. 

Kind regards, 

on behalf of

Dr. David Iain McBride 

Academic Editor

PLOS ONE